# Q-Learning for Continuous Actions with Cross-Entropy Guided Policies

Riley Simmons-Edler [1 2 *]   Ben Eisner [2 *]   Eric Mitchell [2 *]   Sebastian Seung [2]   Daniel Lee [2]

## Abstract

Off-Policy reinforcement learning (RL) is an important class of methods for many problem domains, such as robotics, where the cost of collecting data is high and on-policy methods are consequently intractable. Standard methods for applying Q-learning to continuous-valued action domains involve iteratively sampling the Q-function to find a good action (e.g. via hill-climbing), or by learning a policy network at the same time as the Q-function (e.g. DDPG). Both approaches make tradeoffs between stability, speed, and accuracy. We propose a novel approach, called Cross-Entropy Guided Policies, or CGP, that draws inspiration from both classes of techniques. CGP aims to combine the stability and performance of iterative sampling policies with the low computational cost of a policy network. Our approach trains the Q-function using iterative sampling with the Cross-Entropy Method (CEM), while training a policy network to imitate CEM's sampling behavior. We demonstrate that our method is more stable to train than state of the art policy network methods, while preserving equivalent inference time compute costs, and achieving competitive total reward on standard benchmarks.

## 1. Introduction

In recent years, model-free deep reinforcement learning (RL) algorithms have demonstrated the capacity to learn sophisticated behavior in complex environments. Starting with Deep Q-Networks (DQN) achieving human-level performance on Atari games (Mnih et al., 2013), deep RL has led to impressive results in several classes of challenging tasks. While many deep RL methods were initially limited

to discrete action spaces, there has since been substantial interest in applying deep RL to continuous action domains. In particular, deep RL has increasingly been studied for use in continuous control problems, both in simulated environments and on robotic systems in the real world.

A number of challenges exist for practical control tasks such as robotics. For tasks involving a physical robot where on-robot training is desired, the physical constraints of robotic data collection render data acquisition costly and time-consuming. Thus, the use of off-policy methods like Q-learning is a practical necessity, as data collected during development or by human demonstrators can be used to train the final system, and data can be re-used during training. However, even when using off-policy Q-learning methods for continuous control, several other challenges remain. In particular, training stability across random seeds, hyperparameter sensitivity, and runtime are all challenges that are both relatively understudied and are critically important for practical use.

Inconsistency across runs, e.g. due to different random initializations, is a major issue in many domains of deep RL, as it makes it difficult to debug and evaluate an RL system. Deep Deterministic Policy Gradients (DDPG), a popular off-policy Q-learning method (Lillicrap et al., 2015), has been repeatedly characterized as unstable (Duan et al., 2016; Islam et al., 2017). While some recent work has improved stability in off-policy Q-learning (Haarnoja et al., 2017; 2018c; Fujimoto et al., 2018), there remains significant room for improvement. Sensitivity to hyperparameters (i.e. batch size, network architecture, learning rate, etc) is a particularly critical issue when system evaluation is expensive, since debugging and task-specific tuning are difficult and time consuming to perform. Finally, many real robotics tasks have strict runtime and hardware constraints (i.e. interacting with a dynamic system), and any RL control method applied to these tasks must be fast enough to compute in real time.

Mitigating these challenges is thus an important step in making deep RL practical for continuous control. In this paper, we introduce Cross-Entropy Guided Policy (CGP) learning, a general Q-function and policy training method that can be combined with most deep Q-learning methods and demonstrates improved stability of training across runs, hyperparameter combinations, and tasks, while avoiding

---

[*] Equal contribution [1] Department of Computer Science, Princeton University, Princeton, NJ, USA [2] Samsung Research - AI Center, New York, NY, USA. Correspondence to: Riley Simmons-Edler <rileys@cs.princeton.edu>, Ben Eisner <ben.eisner@samsung.com>.

*Reinforcement Learning for Real Life (RL4RealLife) Workshop in the $36^{th}$ International Conference on Machine Learning*, Long Beach, California, USA, 2019. Copyright 2019 by the author(s).

the computational expense of a sample-based policy at inference time. CGP is a multi-stage algorithm that learns a Q-function using a heuristic Cross-Entropy Method (CEM) sampling policy to sample actions, while training a deterministic neural network policy in parallel to imitate the CEM policy. This learned policy is then used at inference time for fast and precise evaluation without expensive sample iteration. We show that this method achieves performance comparable to state-of-the-art methods on standard continuous-control benchmark tasks, while being more robust to hyperparameter tuning and displaying lower variance across training runs. Further, we show that its inference-time runtime complexity is 3-6 times better than when using the CEM policy for inference, while slightly outperforming the CEM policy. This combination of attributes (reliable training and cheap inference) makes CGP well suited for real-world robotics tasks and other time/compute sensitive applications.

## 2. Related Work

The challenge of reinforcement learning in continuous action spaces has been long studied (Silver et al., 2014; Hafner & Riedmiller, 2011), with recent work building upon *on-policy* policy gradient methods (Sutton et al., 1999) as well as the *off-policy* deterministic policy gradients algorithm (Silver et al., 2014). In addition to classic policy gradient algorithms such as REINFORCE (Sutton et al., 1999) or Advantage Actor Critic (de la Cruz et al., 2018), a number of recent on-policy methods such as TRPO (Schulman et al., 2015) and PPO (Schulman et al., 2017) have been applied successfully in continuous-action domains, but their poor sample complexity makes them unsuitable for many real world applications, such as robotic control, where data collection is expensive and complex. While several recent works (Matas et al., 2018; Zhu et al., 2018; Andrychowicz et al., 2017) have successfully used simulation-to-real transfer to train in simulations where data collection is cheap, this process remains highly application-specific, and is difficult to use for more complex tasks.

Off-policy Q-learning methods have been proposed as a more data efficient alternative, typified by Deep Deterministic Policy Gradients (DDPG) (Lillicrap et al., 2015). DDPG trains a Q-function similar to (Mnih et al., 2016), while in parallel training a deterministic policy function to sample good actions from the Q-function. Exploration is then achieved by sampling actions in a noisy way during policy rollouts, followed by off-policy training of both Q-function and policy from a replay buffer. While DDPG has been used to learn non-trivial policies on many tasks and benchmarks (Lillicrap et al., 2015), the algorithm is known to be sensitive to hyperparameter tuning and to have relatively high variance between different random seeds for a given configu-

ration (Duan et al., 2016; Henderson et al., 2018). Recently multiple extensions to DDPG have been proposed to improve performance, most notably Twin Delayed Deep Deterministic Policy Gradients (TD3) (Fujimoto et al., 2018) and Soft Q-Learning (SQL)/Soft Actor-Critic (SAC) (Haarnoja et al., 2017; 2018b).

TD3 proposes several additions to the DDPG algorithm to reduce function approximation error: it adds a second Q-function to prevent over-estimation bias from being propagated through the target Q-values and injects noise into the target actions used for Q-function bootstrapping to improve Q-function smoothness. The resulting algorithm achieves significantly improved performance relative to DDPG, and we use their improvements to the Q-function training algorithm as a baseline for CGP.

In parallel with TD3, (Haarnoja et al., 2018b) proposed Soft Actor Critic as a way of improving on DDPG's robustness and performance by using an entropy term to regularize the Q-function and the reparametrization trick to stochastically sample the Q-function, as opposed to DDPG and TD3's deterministic policy. SAC and the closely related Soft Q-Learning (SQL) (Haarnoja et al., 2017) have been applied successfully for real-world robotics tasks (Haarnoja et al., 2018c;a).

Several other recent works propose methods that use CEM and stochastic sampling in RL. Evolutionary algorithms take a purely sample-based approach to fitting a policy, including fitting the weights of neural networks, such as in (Salimans et al., 2017), and can be very stable to train, but suffer from very high computational cost to train. Evolutionary Reinforcement Learning (ERL) (Khadka & Tumer, 2018) combines evolutionary and RL algorithms to stabilize RL training. CEM-RL (Pourchot & Sigaud, 2019) uses CEM to sample populations of policies which seek to optimize actions for a Q-function trained via RL, while we optimize the Q-function actions directly via CEM sampling similar to Qt-Opt (Kalashnikov et al., 2018).

There exists other recent work that aims to treat learning a policy as supervised learning (Abdolmaleki et al., 2018b;a; Wirth et al., 2016). Abdolmaleki et al. propose a formulation of policy iteration that samples actions from a stochastic learned policy, then defines a locally optimized action probability distribution based on Q-function evaluations, which is used as a target for the policy to learn (Abdolmaleki et al., 2018b;a).

The baseline for our method is modeled after the CEM method used in the Qt-Opt system, a method described by (Kalashnikov et al., 2018) for vision-based dynamic manipulation trained mostly off-policy on real robot data. Qt-Opt eschews the use of a policy network as in most other continuous-action Q-learning methods, and instead uses

CEM to directly sample actions that are optimal with respect to the Q-function for both inference rollouts and training. They describe the method as being stable to train, particularly on off-policy data, and demonstrate its usefulness on a challenging robotics task, but do not report its performance on standard benchmark tasks or against other RL methods for continuous control. We base our CEM sampling of optimal actions on their work, generalized to MuJoCo benchmark tasks, and extend it by learning a deterministic policy for use at inference time to improve performance and computational complexity, avoiding the major drawback of the method- the need to perform expensive CEM sampling for every action at inference time (which must be performed in real time on robotic hardware).

## 3. Notation and Background

We describe here the notation of our RL task, based on the notation defined by Sutton and Barto (Sutton & Barto, 1998). Reinforcement learning is a class of algorithms for solving Markov Decision Problems (MDPs), typically phrased in the finite time horizon case as an agent characterized by a policy $\pi$ taking actions $a_t$ in an environment, with the objective of maximizing the expected total reward value $\mathbb{E}\sum_{t=1}^{T}\gamma^t r(s_t, a_t)$ that agent receives over timesteps $t \in \{1 \ldots T\}$ with some time decay factor per timestep $\gamma$. To achieve this, we thus seek to find an optimal policy $\pi^*$ that maximizes the following function:

$$J(\pi) = \mathbb{E}_{s,a \sim \pi}[\sum_{t=1}^{T}\gamma^t r(s_t, a_t)]$$

A popular class of algorithms for solving this is Q-learning, which attempts to find an optimal policy by finding a function

$$Q^*(s_t, a_t) = r(s_t, a_t) + \gamma \max_{a_{t+1}}(Q^*(s_{t+1}, a_{t+1}))$$

which satisfies the Bellman equation (Sutton & Barto, 1998):

$$Q(s, a) = r(s, a) + \mathbb{E}[Q(s', a')], \quad a' \sim \pi^*(s')$$

Once $Q^*$ is known $\pi^*$ can easily be defined as $\pi^*(s) = \text{argmax}_a(Q^*(s, a))$. Q-learning attempts to learn a function $Q_\theta$ that converges to $Q^*$, where $\theta$ is the parameters to a neural network. $Q_\theta$ is often learned through bootstrapping, wherein we seek to minimize the function

$$J(\theta) = \mathbb{E}_{s,a}[(Q_\theta - [r(s, a) + \gamma \max_{a'}(\hat{Q}(s', a'))])^2]$$

where $\hat{Q}$ is a target Q-function, here assumed to be a time delayed version of the current Q-function, $\hat{Q}_{\hat{\theta}}$(Mnih et al., 2016).

To use the above equation, it is necessary to define a function $\pi(s)$ which computes $\text{argmax}_a(Q(s, a))$. In discrete

**Algorithm 1** Cross Entropy Method Policy ($\pi_{\text{CEM}}$) for Q-Learning

---
**Input:** state $s$, Q-function $Q$, iterations $N$, samples $n$, winners $k$, action dimension $d$
$\boldsymbol{\mu} \leftarrow \mathbf{0}^d$
$\boldsymbol{\sigma}^2 \leftarrow \mathbf{1}^d$
**for** $t = 1$ **to** $N$ **do**
    $A \leftarrow \{\boldsymbol{a_i} : \boldsymbol{a_i} \overset{\text{i.i.d.}}{\sim} \mathcal{N}(\boldsymbol{\mu}, \boldsymbol{\sigma}^2)\}$
    $\tilde{A} \leftarrow \{\boldsymbol{\tilde{a}_i} : \boldsymbol{\tilde{a}_i} = \tanh(\boldsymbol{a_i})\}$
    $\mathcal{Q} \leftarrow \{q_i : q_i = Q(\boldsymbol{\tilde{a}_i})\}$
    $I \leftarrow \{\text{sort}(\mathcal{Q})_i : i \in [1, \ldots, k]\}$
    $\boldsymbol{\mu} \leftarrow \frac{1}{k}\sum_{i \in I}\boldsymbol{a_i}$
    $\boldsymbol{\hat{\sigma}^2} \leftarrow \text{Var}_{i \in I}(\boldsymbol{a_i})$
    $\boldsymbol{\sigma}^2 \leftarrow \boldsymbol{\hat{\sigma}^2}$
**end for**
**return** $\boldsymbol{\tilde{a}^*} \in \tilde{A}$ such that $Q(\boldsymbol{\tilde{a}^*}) = \max_{i \in I}Q(\boldsymbol{\tilde{a}_i})$

---

action spaces, $\pi(s)$ is trivial, since $\text{argmax}_a$ can be computed exactly by evaluating each possible $a$ with $Q$. In continuous-valued action spaces, such a computation is intractable. Further, as most neural network Q-functions are highly non-convex, an analytical solution is unlikely to exist. Various approaches to solving this optimization problem have been proposed, which have been shown to work well empirically. (Lillicrap et al., 2015) show that a neural network function for sampling actions that approximately maximize the Q-function can be learned using gradients from the Q-function. This approach forms the basis of much recent work on continuous action space Q-learning.

## 4. From Sampling-based Q-learning to Cross-Entropy Guided Policies (CGP)

In this section, we first describe an established method for using a sampling-based optimizer to optimize inputs to a Q-function which can be used as a policy to train the Q-function via standard Q-learning. We then present two novel methods for training deterministic policies separately from the Q-function.

### 4.1. Q-Learning with Sampling-Based Policies

The basis for our method is the use of a sampling-based optimizer to compute approximately optimal actions with respect to a given $Q$ function and a given state $s$. Formally, we define the policy $\pi_{S_Q}(s) = S_Q(s)$, where $S_Q$ is a sampling-based optimizer that approximates $\text{argmax}_a Q(s, a)$ for action $a$ and state observation $s$. We can then train a Q-function $Q_\theta$ parameterized by the weights of a neural network using standard Q-learning as described in Section 3 to minimize:

$$J(\theta) = \mathbb{E}_{s,a}[(Q_\theta - [r(s, a) + \gamma\hat{Q}(s', \pi_{S_{\hat{Q}_\theta}}(s'))])^2]$$

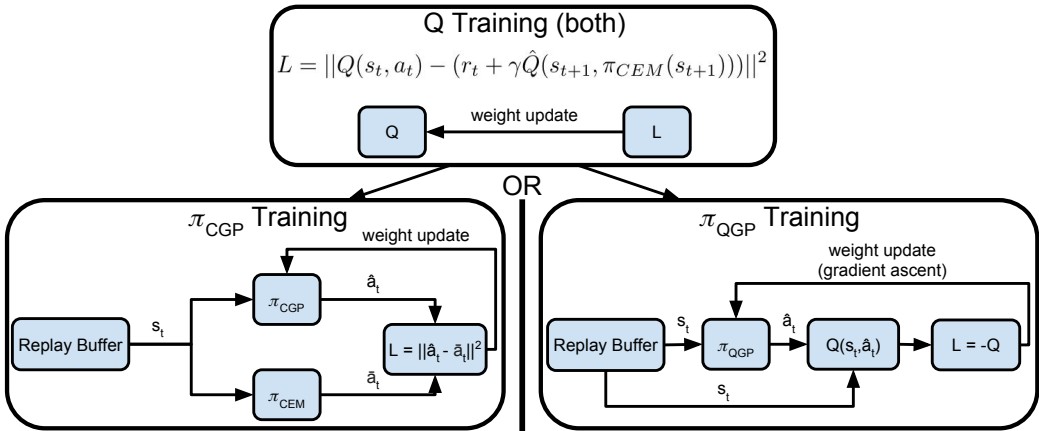

*Figure 1.* Both CGP and QGP utilize the same training method to train their respective Q-functions. However, in CGP (left) we regress $\pi_{CGP}$ on the L2-norm between the current $\pi_{CGP}$ and the CEM-based policy $\pi_{CEM}$. In QGP (right), we train $\pi_{QGP}$ to maximize $Q$ given $s_t$ by directly performing gradient ascent on $Q$.

The choice of sampling-based optimizer $S_Q$ can have a significant impact on the quality of the policy it induces, and therefore has a significant impact on the quality of $Q_\theta$ after training - while we leave the exploration of optimal sampling methods to future work, we used a simple instantiation of the Cross-Entropy method (CEM), which was empirically demonstrated by Kalashnikov et al. to work well for certain continuous-control tasks (Kalashnikov et al., 2018). In this formulation, each action vector is represented as a collection of independent Gaussian distributions, initially with mean $\mu = 1$ and standard deviation $\sigma = 1$. These variables are sampled $n$ times to produce action vectors $a_0, a_1, ..., a_{n-1}$, which are then scored by $Q$. The top $k$ scoring action vectors are then used to reparameterize the Gaussian distributions, and this process is repeated $N$ times. For brevity, we refer to this policy as $\pi_{CEM}$. The full algorithm can be found in Algorithm 1.

### 4.2. Imitating $\pi_{CEM}$ with a Deterministic Policy

While $\pi_{CEM}$ is competitive with learned policies for sampling the Q-function (described in Section 5), it suffers from poor runtime efficiency, as evaluating many sampled actions is computationally expensive, especially for large neural networks. Further, there is no guarantee that sampled actions will lie in a local minimum of the Q-value energy landscape due to stochastic noise. Our main methodological contribution in this work, formalized in Algorithm 2, is the extension of $\pi_{CEM}$ by training a deterministic neural network policy $\pi_\phi(s)$ to predict an approximately optimal action at inference time, while using $\pi_{CEM}$ to sample training data from the environment and to select bootstrap actions for training the Q-function.

A single evaluation of $\pi_\phi$ is much less expensive to compute than the multiple iterations of Q-function evaluations

required by $\pi_{CEM}$. Even when evaluating CEM samples with unbounded parallel compute capacity, the nature of iterative sampling imposes a serial bottleneck that means the theoretical best-case runtime performance of $\pi_{CEM}$ will be $N$ times slower than $\pi_\phi$. Additionally, as $\pi_{CEM}$ is inherently noisy, by training $\pi_\phi$ on many approximately optimal actions from $\pi_{CEM}(s)$ evaluated on states from the replay buffer, we expect that, for a given state $s$ and $Q_\theta$, $\pi_\phi$ will converge to the mean of the samples from $\pi_{CEM}$, reducing policy noise at inference time.

While the idea of training an inference-time policy to predict optimal actions with respect to $Q_\theta$ is simple, there are several plausible methods for training $\pi_\phi$. We explore four related methods for training $\pi_\phi$, the performance of which are discussed in Section 5. The high-level differences between these methods can be found in Figure 1.

#### 4.2.1. Q-GRADIENT-GUIDED POLICY

A straightforward approach to learning $\pi_\phi$ is to use the same objective as DDPG (Lillicrap et al., 2015):

$$J(\phi) = \mathbb{E}_{s \sim \rho^{\pi_{CEM}}} \left( \nabla_{\pi_\phi} Q_\theta(s, \pi_\phi(s)) \right)$$

to optimize the weights $\phi$ off-policy using $Q_\theta$ and the replay data collected by $\pi_{CEM}$. This is the gradient of the policy with respect to the Q-value, and for an optimal Q should converge to an optimal policy. Since the learned policy is not used during the training of the Q-function, but uses gradients from Q to learn an optimal policy, we refer to this configuration as Q-gradient Guided Policies (QGP), and refer to policies trained in this fashion as $\pi_{QGP}$. We tested two versions of this method, an "offline" version where $\pi_\phi$ is trained to convergence on a fixed Q-function and replay buffer, and an "online" version where $\pi_\phi$ is trained in parallel with the Q-function, analogous to DDPG other than

**Algorithm 2** CGP: Cross-Entropy Guided Policies

**TRAINING**
Initialize Q-functions $Q_{\theta_1}, Q_{\theta_2}$ and policy $\pi_\phi$ with random parameters $\theta_1, \theta_2, \phi$, respectively
Initialize target networks $\theta'_1 \leftarrow \theta_1, \theta'_2 \leftarrow \theta_2, \phi' \leftarrow \phi$
Initialize CEM policies $\pi_{\text{CEM}}^{Q_{\theta_1}}, \pi_{\text{CEM}}^{Q_{\theta'_1}}$
Initialize replay buffer $\mathcal{B}$
Define batch size $b$
**for** $e = 1$ **to** $E$ **do**
  **for** $t = 1$ **to** $T$ **do**
    **Step in environment:**
    Observe state $s_t$
    Select action $a_t \sim \pi_{\text{CEM}}^{Q_{\theta_1}}(s_t)$
    Observe reward $r_t$, new state $s_{t+1}$
    Save step $(s_t, a_t, r_t, s_{t+1})$ in $\mathcal{B}$
    **Train on replay buffer ($j \in 1, 2$):**
    Sample minibatch $(s_i, a_i, r_i, s_{i+1})$ of size $b$ from $\mathcal{B}$
    Sample actions $\tilde{a}_{i+1} \sim \pi_{\text{CEM}}^{\theta'_1}$
    Compute $q^* = r_i + \gamma \min_{j \in 1,2} Q_{\theta'_j}(s_{i+1}, \tilde{a}_{i+1})$
    Compute losses $\ell_{Q_j} = \left(Q_{\theta_j}(s_i, a_i) - q^*\right)^2$
    **CGP loss:** $\ell_\pi^{\text{CGP}} = (\pi_\phi(s_i) - \pi_{\text{CEM}}^{\theta_1}(s_i))^2$
    **QGP loss:** $\ell_\pi^{\text{QGP}} = -Q_{\theta_1}(s_i, \pi_\phi(s_i))$
    Update $\theta_j \leftarrow \theta_j - \eta_Q \nabla_{\theta_j} \ell_{Q_j}$
    Update $\phi \leftarrow \phi - \eta_\pi \nabla_\phi \ell_\pi$
    **Update target networks:**
    $\theta'_j \leftarrow \tau \theta_j + (1 - \tau)\theta'_j, \quad j \in 1, 2$
    $\phi' \leftarrow \tau \phi + (1 - \tau)\phi'$
  **end for**
**end for**
**INFERENCE**
**for** $t = 1$ **to** $T$ **do**
  Observe state $s_t$
  Select action $a_t \sim \pi_\phi(s_t)$
  Observe reward $r_t$, new state $s_{t+1}$
**end for**

that $\pi_\phi$ is not used to sample the environment or to select actions for Q-function bootstrap targets. We refer to these variants as QGP-Offline and QGP-Online respectively.

### 4.2.2. CROSS-ENTROPY-GUIDED POLICY

However, as shown in Figure 3, while both variants that train $\pi_\phi$ using the gradient of $Q_\theta$ can achieve good performance, their performance varies significantly depending on hyperparameters, and convergence to an optimal (or even good) policy does not always occur. We hypothesize that the non-convex nature of $Q_\theta$ makes off-policy learning somewhat brittle, particularly in the offline case, where gradient ascent on a static Q-function is prone to overfitting to local maxima. We therefore introduce a second variant, the Cross-Entropy Guided Policy (CGP), which trains $\pi_\phi$ using

an L2 regression objective

$$J(\phi) = \mathbb{E}_{s_t \sim \rho^{\pi_{\text{CEM}}}}(\nabla_{\pi_\phi}||\pi_\phi(s_t) - \pi_{\text{CEM}}(s_t)||^2)$$

This objective trains $\pi_\phi$ to imitate the output of $\pi_{\text{CEM}}$ without relying on CEM for sampling or the availability of $Q_\theta$ at inference time. If we assume $\pi_{\text{CEM}}$ is an approximately optimal policy for a given $Q_\theta$ (an assumption supported by our empirical results in Section 5), this objective should converge to the global maxima of $Q_\theta$, and avoids the local maxima issue seen in QGP. As $\pi_{\text{CEM}}$ can only be an approximately optimal policy, CGP may in theory perform worse than QGP since QGP optimizes $Q_\theta$ directly, but we show that this theoretical gap does not result in diminished performance. Moreover, we demonstrate that CGP is significantly more robust than QGP, especially in the offline case. We explore both online and offline versions of this method similar to those described for QGP.

While QGP and CGP are compatible with any Q-learning algorithm, to improve performance and training stability further we combine them with the TD3 Q-learning objective described in (Fujimoto et al., 2018), which adds a second Q-function for target Q-value computation to minimize function approximation error, among other enhancements. Our method of using $\pi_{\text{CEM}}$ to sample actions for Q-function training and training $\pi_\phi$ for use at inference time is agnostic to the form of the Q-function and how it is trained, and could be combined with future Q-learning methods. Pseudocode for the full CGP method can be found in Algorithm 2.

## 5. Experiments

To characterize our method, we conduct a number of experiments in various simulated environments.

### 5.1. Experiment Setup

Our experiments are intended to highlight differences between the performance of CGP and current state-of-the-art methods on standard RL benchmarks. We compare against DDPG, TD3, Soft Actor-Critic (SAC), and an ablation of our method which does not train a deterministic policy but instead simply uses $\pi_{\text{CEM}}$ to sample at test time similar to the method of (Kalashnikov et al., 2018). To obtain consistency across methods and with prior work we used the author's publicly available implementations for TD3 and SAC, but within our own training framework to ensure consistency. We attempt to characterize the behavior of these methods across multiple dimensions, including maximum final reward achieved given well-tuned hyperparameters, the robustness of the final reward across diverse hyperparameters, the stability of runs within a given hyperparameter set, and the inference time computational complexity of the method.

We assess our method on an array of continuous control tasks in the MuJoCo simulator through the OpenAI gym interface, including HalfCheetah-v2, Humanoid-v2, Ant-v2, Hopper-v2, Pusher-v2 (Brockman et al., 2016). These tasks are intended to provide a range of complexity, the hardest of which require significant computation in order to achieve good results. The dimensionality of the action space ranges from 2 to 17 and the state space 8 to 376. Because of the large amount of computation required to train on these difficult tasks, robustness to hyperparameters is extremely valuable, as the cost to exploring in this space is high. For similar reasons, stability and determinism limit the number of repeated experiments required to achieve an estimate of the performance of an algorithm with a given degree of certainty. In order to test robustness to hyperparameters, we choose one environment (HalfCheetah-v2) and compare CGP with other methods under a sweep across common hyperparameters. To test stability, we perform 4 runs with unique random seeds for each hyperparameter combination. Each task is run for 1 million time steps, with evaluations every 1e4 time steps.

After tuning hyperparameters on HalfCheetah-v2, we then selected a single common "default" configuration that worked well on each method, the results of which for HalfCheetah-v2 are shown in Figure 2. We then ran this configuration on each other benchmark task, as a form of holdout testing to see how well a generic set of hyperparameters will do for unseen tasks.

We also include several variants of our method, as described in Section 4.2. We compare robustness and peak performance for both online and offline versions of CGP and QGP.

## 5.2. Comparisons

**Performance on standard benchmarks**  When run on 5 different standard benchmark tasks for continuous-valued action space learning (HalfCheetah-v2, Humanoid-v2, Hopper-v2, Pusher-v2, and Ant-v2), CGP achieves maximum reward competitive with the best methods on that task (with the exception of Ant-v2, where TD3 is a clear winner). Importantly, CGP performed consistently well across all tasks, even though its hyperparameters were only optimized on one task- in all tasks it is either the best or second best method. Other methods (i.e. SAC and TD3) perform well on one task with the given set of hyperparameters, such as TD3 on Ant-v2 or SAC on Humanoid-v2, but perform poorly on one or more other tasks, as TD3 performs poorly on Humanoid-v2 and SAC on Ant-v2. We note that for each method better performance can be achieved using hyperparameters tuned for the task (for example, Haarnoja et al. report much better performance on Humanoid-v2 using task-specific hyperparameters (Haarnoja et al., 2018d)),

but as we are interested in inter-task hyperparameter robustness we do not perform such tuning. Additionally, even though CGP is based on the Q-function used in the TD3 method, it greatly outperforms TD3 on complex tasks such as Humanoid-v2, suggesting that the CEM policy is more robust across tasks. See Figure 2 for details.

**Stability across runs**  Across a wide range of hyperparameters (excluding very large learning rate $\geq 0.01$), CGP offers a tight clustering of final evaluation rewards. Other methods demonstrated higher-variance results, where individual runs with slightly different hyperparameters would return significantly different run distributions. To arrive at this conclusion, we ran a large battery of hyperparameter sweeps across methods, the detailed results of which can be observed in Appendix A of the supplement. We consider CGP's relative invariance to hyperparameters that are sub-optimal one of its most valuable attributes; we hope that it can be applied to new problems with little or no hyperparameter tuning.

**Robustness across hyperparameters**  We evaluated the robustness of each method over hyperparameter space, using a common set of hyperparameter configurations (with small differences for specific methods based on the method). For most hyperparameters, we held all others fixed while varying only that parameter. We varied learning rate (LR) and batch size jointly, with smaller learning rates matching with smaller batch sizes, and vice versa. We varied LR among the set $\{0.01, 0.001, 0.0001\}$, and batch size among $\{256, 128, 64, 32\}$. We also independently varied the size of the network in $\{512, 256, 128, 32\}$. For methods using random sampling for some number of initial timesteps (CGP and TD3), we varied the number in $\{0, 1000, 10000\}$, and for those which inject noise (all other than SAC) we varied the exploration and (for CGP and TD3) next action noise in $\{0.05, 0.1, 0.2, 0.3\}$. We evaluated CGP entirely with no exploration noise, which other methods using deterministic policies (TD3, DDPG) cannot do while remaining able to learn a non-trivial policy. The overall results of these sweeps can be seen in Figure 3, while detailed results breaking the results down by hyperparameter are in the supplement.

Overall, we see that while CGP does not perform as well in the top quartile of parameter sweeps, it displays a high degree of stability over most hyperparameter combinations, and displays better robustness than SAC in the lower half of the range and DDPG everywhere. CGP also performs slightly better for almost all states than the CEM policy it learns to imitate. The failure cases in the tail were, specifically, too high a learning rate (LR of 0.01, which is a failure case for CGP but not for CEM) and less initial random sampling (both 0 and 1000 produced poor policies for some seeds).

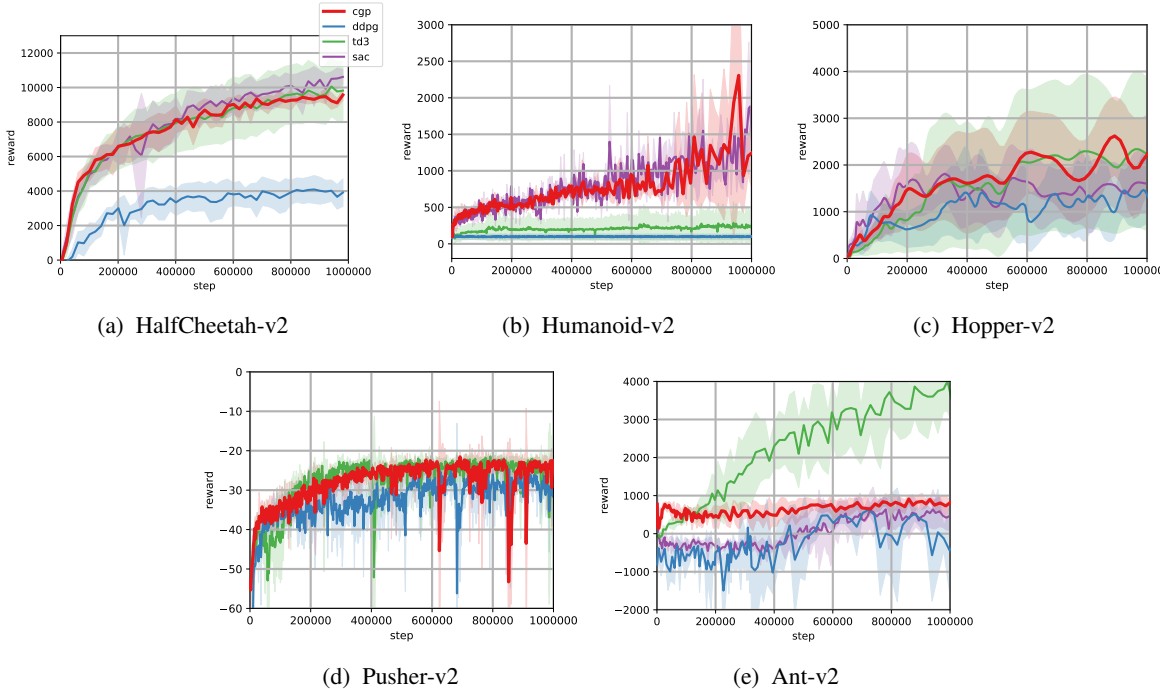

(a) HalfCheetah-v2  (b) Humanoid-v2  (c) Hopper-v2

(d) Pusher-v2  (e) Ant-v2

*Figure 2.* Performance of various methods (CGP, SAC, DDPG, and TD3) on OpenAI Gym benchmark tasks, simulated in a MuJoCo environment. We note that in all cases CGP is either the best or second best performing algorithm, while both TD3 and SAC perform poorly on one or more tasks, and DDPG fails to train stably on most tasks. The thick lines represent the mean performance for a method at step $t$ across 4 runs, and upper and lower respectively represent the max and min across those runs. Parameters for each method were optimized on the HalfCheetah-v2 benchmark, and then applied across all other benchmark tasks. Due to high variance we applied smoothing to all trend lines for all methods for Hopper-v2.

**Inference speed and training efficiency** We benchmarked the training and inference runtime of each method. We computed the mean over 10 complete training and inference episodes for each method with the same parameters. The results can be found in Table 1. CEM-2, CGP-2, CEM-4, and CGP-4 refer to the number of iterations of CEM used(2 or 4). $\pi_{\text{CGP}}$ greatly outperforms $\pi_{\text{CEM}}$ at inference time, and performs the same at training time. Other methods are faster at training time, but run at the same speed at inference time Importantly, the speed of $\pi_{\text{CGP}}$ at inference time is independent of the number of iterations of CEM sampling used for training.

### 5.3. CGP Variants

We consider several variants of our method, as detailed in Section 4.2. We ran each variant on a suite of learning rate and batch size combinations to evaluate their robustness. We tested LR values in $\{0.001, 0.0001\}$ and batch sizes in $\{32, 128, 256\}$. See Figure 4 for a comparison of all runs performed.

**CGP versus QGP** The source of the supervision signal is a critical determinant of the behavior of the policy. Thus it

*Table 1.* Runtime in average seconds per episode of HalfCheetah-v2 (without rendering) on an otherwise-idle machine with a Nvidia GTX 1080 ti GPU. CGP achieves a constant inference runtime independent of the number of CEM iterations used, which matches the performance of other methods.

| METHOD | MEAN TRAIN (S) | MEAN INFERENCE (S) |
|--------|----------------|--------------------|
| RANDOM | - | 0.48 |
| DDPG | 5.75 | 2.32 |
| TD3 | 5.67 | 2.35 |
| SAC | 11.00 | 2.35 |
| CEM-2 | 7.1 | 6.3 |
| CEM-4 | 9.3 | 10.1 |
| CGP-2 | 11.03 | 2.35 |
| CGP-4 | 14.46 | 2.35 |

is important to compare the performance of the policy when trained to directly optimize the learned Q-function and when trained to imitate CEM inference on that same policy. We find that directly optimizing the learned Q-function suffers from more instability and sensitivity to chosen hyperparameters, particularly when learning offline. In comparison, both CGP variants train well in most cases. This suggests that the CEM approximation to the policy gradient is not only a

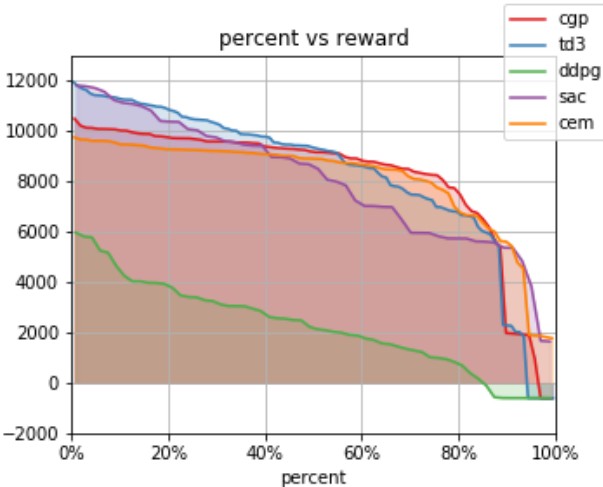

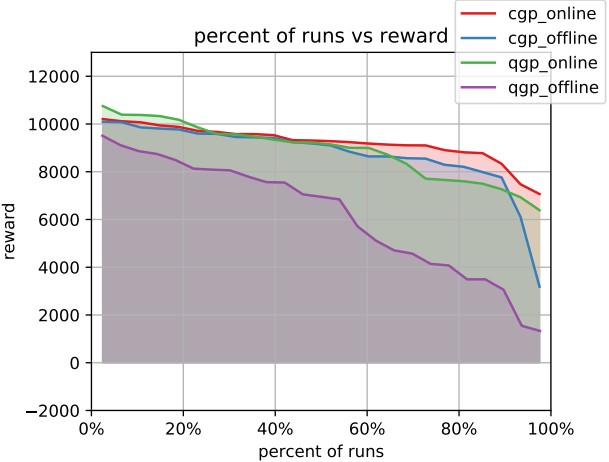

*Figure 3.* Stability of various methods on the HalfCheetah-v2 benchmark task. This figure shows the percentage of runs across all hyperparameter configurations that reached at least the indicated reward level. Each hyperparameter set was run for 1000 episodes, with 4 replicates for each run. The same hyperparameter sets were used across methods. While CGP is outperformed with optimized parameters, its performance decays much slower for sub-optimal configurations.

*Figure 4.* Stability of variants of CGP, as measured by the percent of runs across all hyperparameter configurations tested reaching at least the total reward given on the y-axis. Three of the four methods (CGP-Online, CGP-Offline, and QGP-Online) performed roughly equivalently in the upper quartiles, while CGP-Online performed better in the bottom quartile.

reasonable approximation but is also easier to optimize than the original gradient.

**Online versus Offline** Another dimension of customization of CGP is the policy training mode; training can either be online (train the policy function alongside the Q-function, with one policy gradient step per Q-function gradient step) or offline (train only at the end of the Q-function training trajectory). An advantage of the CGP method is that it performs similarly in both paradigms; thus, it is suitable for completely offline training when desired and online learning when the Q-function is available during training.

We find that the online training runs of both CGP and QGP are mildly better than offline training. This result is somewhat intuitive if one considers the implicit regularization provided by learning to optimize a non-stationary Q-function, rather than a static function, as in the offline learning case. Ultimately, CGP is effective in either regime.

## 6. Discussion

In this work, we presented Cross-Entropy Guided Policies (CGP) for continuous-action Q-learning. We show that CGP is robust and stable across hyperparameters and random seeds, competitive in performance when compared with state of the art methods, as well as both faster and more accurate than the underlying CEM policy. We demonstrate that not only is CEM an effective and robust general-purpose optimization method in the context of Q-learning, it is an ef-

fective supervision signal for a policy, which can be trained either online or offline. Our findings support the conventional wisdom that CEM is a particularly flexible method for reasonably low-dimensional problems (Rubinstein, 1997), and our findings suggest that CEM remains effective even for problems that have potentially high-dimensional latent states, such as Q-learning.

We would also like to consider more of the rich existing analysis of CEM's properties in future work, as well as explore other sample-based algorithms for optimizing the actions of Q-functions, such as covariance matrix adaptation (Hansen & Ostermeier, 2001). Another direction to explore is entropy-based regularization of the Q-function similar to SAC (Haarnoja et al., 2018d), which may further improve stability and make the Q-function easier to optimize, as an entropy objective encourages Q-value smoothness.

We believe that there is potential for further gains in stable and robust continuous action Q-learning through sampling methods. While such developments may come at a computational cost, our success in training inference-time policies shows that by doing so we achieve runtime performance comparable to other non-sampling methods independent of sampling compute times. Therefore, we believe that sample-based Q-function optimization represents a promising new direction for continuous-action Q-learning research that offers unique advantages and can combine well with other Q-learning methods.

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
