# OpenReview forum: "Q-Learning for Continuous Actions with Cross-Entropy Guided Policies"
_ICML.cc/2019/Workshop/RL4RealLife — RL4RealLife 2019_

### Official Review · AnonReviewer2 · 2019-05-23
**This paper propose a new approach of iterative sampling policy called cross entropy guided policy(CGP) and compared its performance with Q-gradient guided policy(QGP). The author claimed proposed CGP outperforms in its stability and performance.**

**Rating:** 4
**Confidence:** 2

**Review:**

Pros:
- This paper proposes novel methods with rigorous and detail theoretical explanations of the method.
- The paper is well written and the proposed method is clearly explained with proper notation and formulations.
- The author compared proposed method (CGP) with existing method (QGP) through intensive experiments and shown results are promising.
- Their results are convincing and they demonstrated that their method outperforms over existing method in its stability on standard benchmarks.

Cons:
The topic of this paper is theoretical. Therefore, it is a bit off from the main topic of this workshop which is mainly focusing on application of RL for real life problem.

---

### Official Review · AnonReviewer1 · 2019-05-24
**Not a generic or novel approach**

**Rating:** 3
**Confidence:** 4

**Review:**

In this paper, the authors study RL in MDPs where the action space is continuous.  Continuous action space problems are prominent in the control task. The authors nicely motivate this task and describe the prior works on on-policy approaches. They argue that on-policy methods might not be sample efficient, therefore off-policy methods might be in interest. They argue that the current state of off-policy and on-policy methods for continuous tasks are sensitive to hyperparameter, initialization and are unstable.  They dedicate their paper to study this problem and propose a method to remedy this issue.

The paper is well written, but there is not much content in it. It could have been much shorter.

The authors first study an idea similar to DDPG, where instead of using a Q(x,\pi(a)) to compute the policy gradient on \pi, they use a native sampling approach based on cross-entropy method.
This approach has been previously proposed by Kalashnikov et al., 2018. Following this approach, at each time step, we draw a few actions and ranks them based on their Q function. We keep the best ones and then iteratively redo the sample based on the chosen ones. This approach is expected to resemble argmax_a Q(x,a), therefore, max_a Q(x,a)\sim Q(x,\pi_cem(x))
Following this approach, we collect experience using policy \pi_cem and use Q(x,\pi(a)) to update the policy.

The authors suggest to instead of following this approach or approaches similar to DDPG that implement policy gradient using samples generated by \pi_cem, totally throw away the Q estimation and directly imitate \pi_cem by minimizing ||\pi_cem-\pi||
While this idea is not novel or compelling, the experimental study is also not compelling. I doubt this approach would be helpful beyond the special cases. At least it is not clear why it should be the case.

An important point on frequents misuse of term Q-learning:
The authors clearly provided the definition of Q-learning at the beginning of section 3.  But, throughout this paper, they have misused this term multiple time. If we minimize L2 losses like section 4.1, that algorithm is not the Q-learning algorithm.
Similarly,  most of the deep RL algorithms that the authors referred them as Q-learning, are not Q-learning.
Please consider rephrasing.

---

### Decision · Program_Chairs · 2019-05-28

Accept